# The Clinicopathological Spectrum of Acromegaly

**DOI:** 10.3390/jcm8111962

**Published:** 2019-11-13

**Authors:** Amit Akirov, Sylvia L. Asa, Lama Amer, Ilan Shimon, Shereen Ezzat

**Affiliations:** 1Department of Endocrine Oncology, Princess Margaret Cancer Centre, Toronto, ON M5G 2M9, Canada; Lama.amer@uhn.ca (L.A.); Shereen.Ezzat@sinaihealthsystem.ca (S.E.); 2Institute of Endocrinology, Beilinson Hospital, 49100 Petach Tikva, Israel; i_shimon@netvision.net.il; 3Sackler School of Medicine, Tel Aviv University, 6997801 Tel Aviv, Israel; 4Department of Pathology, University Hospitals, Cleveland, Case Western Reserve University, Cleveland, OH 44106, USA; pathlady01@gmail.com

**Keywords:** acromegaly, pituitary tumor, somatotroph tumor, ectopic hormone production

## Abstract

Background: Acromegaly results from a persistent excess in growth hormone with clinical features that may be subtle or severe. The most common cause of acromegaly is a pituitary tumor that causes excessive production of growth hormone (GH), and rare cases are due to an excess of the GH-releasing hormone (GHRH) or the ectopic production of GH. Objective: Discuss the different diseases that present with manifestations of GH excess and clinical acromegaly, emphasizing the distinct clinical and radiological characteristics of the different pathological entities. Methods: We performed a narrative review of the published clinicopathological information about acromegaly. An English-language search for relevant studies was conducted on PubMed from inception to 1 August 2019. The reference lists of relevant studies were also reviewed. Results: Pituitary tumors that cause GH excess have several variants, including pure somatotroph tumors that can be densely or sparsely granulated, or plurihormonal tumors that include mammosomatotroph, mixed somatotroph-lactotroph tumors and mature plurihomonal Pit1-lineage tumors, acidophil stem cell tumors and poorly-differentiated Pit1-lineage tumors. Each tumor type has a distinct pathophysiology, resulting in variations in clinical manifestations, imaging and responses to therapies. Conclusion: Detailed clinicopathological information will be useful in the era of precision medicine, in which physicians tailor the correct treatment modality to each patient.

## 1. Introduction

Acromegaly results from persistent excess growth hormone (GH), which through the GH receptor stimulates the synthesis of circulating insulin-like growth factor-1 (IGF-1) mainly in the liver. High levels of IGF-1 promote cell proliferation and inhibit apoptosis, and are responsible for most of the clinical manifestations of acromegaly [1,2].

The clinical features of acromegaly may be subtle or severe, and range from acral overgrowth, soft-tissue swelling, joint pain, jaw prognathism and hyperhidrosis to frontal bone bossing, diabetes mellitus, hypertension, respiratory and cardiac failure [2]. Local tumor effects, visceromegaly and reproductive dysfunction are also common. Furthermore, accelerated growth and gigantism may develop when the tumor arises in young patients before the closure of the epiphyseal bone [2].

The most common cause of acromegaly is a pituitary tumor that causes excessive production of GH. Ectopic production of GH is exceptionally uncommon; rare cases of pancreatic neuroendocrine tumor or lymphoma have been reported as an extrapituitary source of GH excess.

In contrast, excess production of the GH-releasing hormone (GHRH) is an unusual but well documented cause of acromegaly, and may be associated with neuroendocrine tumors of lung, pancreas, thyroid (medullary thyroid cancer) or pheochromocytomas [2,3], as well as hypothalamic gangliocytomas [4,5].

While most cases are sporadic, there are several familial syndromes, such as multiple endocrine neoplasia type 1 (MEN1) and 4 (MEN4), familial isolated pituitary adenoma (FIPA) and Carney complex, as well as the sporadic germline mosaic disorder McCune-Albright disease, that predispose to pituitary hyperplasia and neoplasia, causing acromegaly or gigantism [6,7]. In these cases the presentation can be quite severe, with its onset at a young age, high GH levels and poor response to medical treatment [8]. A rare genetic syndrome, X-linked acrogigantism (X-LAG), has also been recently implicated in early onset childhood gigantism [9].

Pituitary tumors that cause GH excess have several variants, including pure somatotroph tumors that can be densely-granulated or sparsely granulated, or plurihormonal tumors that include mammosomatotroph, mixed somatotroph-lactotroph tumors and mature plurihomonal Pit1-lineage tumors, as well as the rare acidophil stem cell tumors and poorly-differentiated Pit1-lineage tumors. This complex family of neoplasms can express multiple hormones, including GH, prolactin, β-thyroid stimulating hormone and/or α-subunit of glycoprotein hormones [10]. Clinicopathological studies have shown that each tumor type has a distinct pathophysiology, resulting in variations in clinical manifestations, imaging and responses to therapies [11]. In addition, acromegaly caused by pituitary hyperplasia, either primary or due to ectopic GHRH, also has unique clinical and radiologic features, and may require a completely different therapeutic approach.

In this review we will discuss the different diseases that present with manifestations of GH excess and clinical acromegaly or gigantism, emphasizing the distinct clinical and radiological characteristics of each pathogenetic entity and the various approaches to management.

## 2. Primary Pituitary Causes

The pituitary is usually the primary site of pathology. The disease may be due to a neoplasm or to primary hyperplasia. There are many types of pituitary neuroendocrine tumors that can cause acromegaly, and each has a distinct pathophysiology.

### 2.1. Densely-granulated Somatotroph Tumor

Densely-granulated somatotroph tumors, found in 30–50% of acromegaly patients, are composed of deeply eosinophilic tumor cells (Figure 1), with a diffuse positivity for GH and a perinuclear staining pattern of low molecular weight keratins, that closely resemble normal somatotrophs [12,13,14]. Like normal somatotrophs, they express the α-subunit of glycoprotein hormones. Electron microscopy shows a well-developed rough endoplasmic reticulum, large Golgi complex, numerous perinuclear intermediate filaments and numerous large (300–600 nm), round, electron dense secretory granules that contain GH [10].

These tumors usually present in patients older than 50 years and are usually slow growing lesions. Several studies have reported that densely-granulated somatotroph tumors are frequently associated with high levels of GH and IGF-1, as well as a florid and symptomatic presentation of acromegaly [15,16,17,18,19].

The clinical diagnosis of a densely-granulated somatotroph tumor can be predicted by radiologic imaging (Figure 2). Indeed, features of acromegaly that can be seen on the image include frontal bossing, prognathism and thickening of the scalp with a “rug sign”. T2-weighted MRI signal intensity is a marker for granulation; densely-granulated tumors are associated with a low T2 intensity compared with sparsely-granulated tumors [20].

The biochemical response to treatment with somatostatin analogs is quite high in patients with densely-granulated tumors, ranging between 65–90% [16,21,22,23]. The high response rate is possibly related to the frequently elevated cyclic adenosine monophosphate (cAMP) levels in these tumors that harbor activating mutations of *GNAS* which cause constitutive activation of Gsα [10,24,25,26,27]. This also likely explains the expression of an α-subunit by these tumors, since that gene has a CREB binding site [27].

### 2.2. Sparsely-Granulated Somatotroph Tumor

Sparsely-granulated somatotroph tumors, found in 15–35% of patients with acromegaly, are composed of lightly eosinophilic or chromophobic cells, that are characterized by the presence of conspicuous juxtanuclear keratin aggregates known as “fibrous bodies” (Figure 3). These tumors have weak or focal positivity for GH and do not express the α-subunit [10,13,14]. They also have a marked reduction of E-cadherin [28], a feature that explains the frequent dehiscence of their architecture. Electron microscopy demonstrates large aggregates of keratin filaments that form a whorled structure in the juxtanuclear region. These structures trap the Golgi complex and may also contain scattered small secretory granules that contain GH [10]. A variant of somatotroph tumors, that may have occasional “fibrous bodies”, is an intermediate type of tumor [29] but behaves clinically like a densely-granulated somatotroph tumor rather than the sparsely-granulated tumor that is very distinctive, and therefore is not classified in this category [10]. Sparsely-granulated tumors can be more aggressive, with Ki67 proliferation indices >3% in most cases [30].

Sparsely-granulated somatotroph tumors are more common in patients who are younger than 50 years of age; they often present with a more rapidly growing tumor and are larger at diagnosis, compared to densely-granulated tumors [16,17,19,29,31,32]. The clinical manifestations of acromegaly may be more subtle in these patients where the hormonal syndrome may be missed and the tumor misclassified as clinically “silent” [33]. Levels of GH and IGF-1 might not be as high as in patients with densely-granulated somatotroph tumors [16,17,19].

Sparsely-granulated tumors are associated with a characteristic T2-hyperintensity on the MRI [15,34] (Figure 4), and therefore the diagnosis can often be predicted based on imaging. Increased incidence of cavernous sinus invasion, compared to densely-granulated tumors, was reported by some studies [18,19,29], but not in others [16,17].

Importantly, these tumors are often resistant to treatment with somatostatin analogs [21,22,23]. This explains why multiple studies have found that low T2-weighted intensity predicts a higher response rate to somatostatin analogs therapy, compared to higher T2 intensity tumors [15,16,18,19,34,35,36].

The pathogenesis of these tumors is clearly distinct from that of densely-granulated somatotroph tumors. In young patients with acromegaly/gigantism onset <30 years of age and a sparsely-granulated somatotroph tumor, especially in the presence of a positive family history, an investigation for germline line mutation of the aryl hydrocarbon receptor-interacting protein (AIP), causing familial isolated pituitary adenoma (FIPA), should be considered [37]. There is evidence of epigenetic silencing of AIP with reduced expression in sparsely-granulated somatotroph tumors that are not associated with FIPA [38]. One study reported that sparsely-granulated tumors have a somatic mutation in the GH-receptor gene that alters GH autoregulation [39]. The lack of expression of an α-subunit by these tumors emphasizes the lack of high cAMP levels. However, Larkin et al. reported that the granulation pattern, but not the genotype, corresponds to clinical and biochemical characteristics and the response to SSA treatment, and in their study mutations in GH-receptor or GSP oncogenes were unrelated to the granulation pattern [19].

Sparsely-granulated tumors have been reported to have lower SSTR2 expression; this has been implicated as the cause of a lower response rate to somatostatin analogs, compared to densely-granulated tumors [21]. Kiseljak-Vassiliades et al. identified alterations in the expression of cell cycle kinase inhibitor p27^Kip1^, which was lower in sparsely- vs. densely-granulated tumors, and showed association between E-cadherin upregulation with increased p27^Kip1^ levels. These alterations may also partially explain the relative resistance of sparsely-granulated tumors to somatostatin analogs [40]. However, one study reported that the biochemical and clinical response was greater in patients with sparsely- compared with densely-granulated tumors when pasireotide was used, following the failure of first-generation somatostatin analogs [41]. This greater response to pasireotide, a multi-receptor targeted somatostatin analog that binds SSTR1, SSTR2 and SSTR3, and has the highest affinity for SSTR5, may be related to the higher expression of SSTR5 in sparsely-granulated tumors [42,43].

The literature has been inconsistent regarding the surgical response based on GH tumor subtype. While Mazal et al. reported a higher rate of incomplete resection and additional surgical interventions in patients with sparsely-granulated tumors [16,44], others did not find significant difference between sparsely- and densely-granulated tumors [17]. There are conflicting reports regarding significant differences in tumor recurrence rates between sparsely- and densely-granulated tumors, as several studies having found no difference between the two types [17], while others reported higher recurrence rates with sparsely-granulated tumors [16]. These distinctions are based on the surgical resectability of the lesions; while sparsely-granulated tumors tend to be larger and more invasive, even densely-granulated tumors are usually diagnosed late, as the average onset of symptoms is usually more than 10 years prior to diagnosis, and most are already invasive, resulting in incomplete resection [45].

One report regarding the response to treatment with pegvisomant, a GH receptor antagonist, showed that unlike the significant difference in response to somatostatin analogs, there was no difference in response to pegvisomant between sparsely-granulated and densely-granulated tumors, as all patients with sparsely-granulated tumors showed normalization of IGF-1 levels while on this treatment [16]. This is not surprising, since the drug acts peripherally on the GH receptor, not the pituitary tumor itself.

Lee et al. reported similar response rate to stereotactic radiosurgery in patients with densely- or sparsely-granulated tumors, ranging between 70% and 80% within four years of therapy [46]. Similarly, patients with densely- or sparsely-granulated tumors developed new pituitary hormone deficiency after radiotherapy in similar rates (range 40–70% after 6–8 years of therapy) [46].

### 2.3. Mammosomatotroph Tumor

Mammosomatotroph tumors are composed of a single monomorphous population of Pit1-lineage cells that express both GH and prolactin, and also are positive for the alpha-subunit. These cells are strongly acidophilic, and resemble densely-granulated somatotrophs histologically and immunohistochemically, with the exception of the additional expression of ERα and prolactin (Figure 1) [10]. By electron microscopy they also resemble somatotrophs, but have more variable sizes and shapes of secretory granules that can range from 200–2000 nm, and may show the misplaced exocytoses that are characteristic of lactotrophs [10,13].

The clinical and biological features of mammosomatotroph tumors are very similar to the characteristics of densely-granulated somatotroph tumors with the exception that patients may have more significant hyperprolactinemia [14]. High prolactin levels can be seen in patients with somatotroph tumors that interrupt the pituitary stalk, but the levels are higher in patients with bihormonal tumors (>200 µg/L) [47].

While densely-granulated somatotroph tumors are the most common cause of acromegaly in adults, mammosomatotroph tumors that produce both GH and prolactin are the most frequent tumors in young patients with acromegaly and in cases of childhood onset-gigantism [27]. As mammosomatotroph tumors resemble densely-granulated somatotrophs, they similarly have high secretory activity resulting in florid clinical manifestations, leading to earlier diagnosis when the tumors are relatively small [30]. A recent study reviewed 94 patients with acromegaly and categorized the cases into three groups: pure somatotroph tumor (53 cases), mammosomatotroph (28 cases) and mixed somatotroph-lactotroph (13 cases) tumor. Of the three subtypes, mammosomatotroph tumors often had the smallest size (mean 17.5 cm), lowest frequency of cavernous sinus invasion (7.1%) and gross total tumor resection was most frequent (85.7%) [47].

Because these tumors are densely granulated, they similarly have low intensity on T2-weighted MR imaging.

It is likely that these tumors are also among the lesions with the activation of cyclic AMP. The rationale for this logic is that they have been frequently reported in patients with McCune Albright Syndrome. In this setting, the pituitary may show areas of hyperplasia along with areas of tumor.

Limited data are available in the literature regarding the response to medical treatment of these tumors, and it is likely that similar to densely-granulated tumors, these tumors are responsive to somatostatin analogs, as there is evidence for response within in vitro studies [48], but they may also respond to dopamine agonists.

### 2.4. Mature Plurihormomal Pit1-Lineage Tumor

Rare pituitary tumors that resemble mammosomatotroph tumors may synthesize and secrete TSH and also express the transcription factor GATA3 [49]. These tumors are almost identical to mammosomatotrophs, but the patients may also have hyperthyroidism [10,30].

### 2.5. Mixed Somatotroph-Lactotroph Tumor

These tumors are composed of two distinct cell populations, somatotrophs and lactotrophs. Either cell type can be densely or sparsely granulated, and various combinations may occur: densely- or sparsely-granulated somatotrophs can be admixed with densely- or sparsely-granulated lactotrophs [10]. These are distinct from mammosomatotroph tumors, which are composed of a single monomorphous cell population that expresses both hormones. These tumors express Pit1 in all tumor cells, but only the cells that express prolactin also express ERα [33].

The characteristics of mixed somatotroph and lactotroph tumors are dependent on the specific composition of the tumor cells and the relative proportions of the two cell components [10,14]

The literature regarding the behavior of these tumors is confounded by a lack of clear pathology classification of tumors that produce both GH and prolactin, as these include mammosomatotroph, mixed somatotroph and lactotroph as well as plurihormonal and poorly-differentiated Pit1-lineage tumors that all have a distinct pathogenesis, and in the mixed category, there can be all the different patterns of sparsely- and densely-granulated somatotrophs and lactotrophs [10]. Several studies have reported that mixed tumors have an increased risk of invasion into surrounding structures, are difficult to treat, and have a low surgical cure rate [14,50]. In the study by Lv et al., the mixed somatotroph-lactotroph tumors were characterized by the biggest tumor size and lowest successful tumor resection, compared to mammosomatotroph or pure somatotroph tumors [47]. This may be attributed to the sparsely-granulated somatotroph component in many of these tumors. Rick et al. reported that compared to pure somatotroph tumors, tumors that stain for both GH and prolactin presented with significantly higher serum IGF-1 (803.6 vs. 480.0 µg/L) and prolactin (60.7 vs. 10 µg/L) levels, had a lower remission rate (32% vs. 80%) and the recurrence risk was higher (18.2% vs. 7%), although in their study, the mean tumor size was similar in both groups. Ultimately, remission following surgery was less likely in patients with tumors that stained for both GH and prolactin, and while all patients with single-staining tumors achieved remission with monotherapy, 30% of those with bihormonal tumors required combination treatment for disease control. Furthermore, doses of pegvisomant, cabergoline and somatostatin analogs were higher in patients with bihormonal tumors [51]. However, in this study, there was no distinction between mammosomatotroph tumors, mixed somatotroph-lactotroph tumors and poorly-differentiated Pit1-lineage tumors that all have different biological features, so the value of these data is unclear.

Dopamine agonists such as cabergoline are a possible treatment option in acromegaly, as mammosomatotroph tumors, mixed somatotroph-lactotroph tumors and pure somatotroph tumors have dopamine receptors on their surface, and studies have revealed that even patients without hyperprolactinemia may show significant response to cabergoline [52,53]. In a meta-analysis to investigate the place of cabergoline in acromegaly, Sandret et al. reported IGF-1 normalization with the dopamine agonist in five out of eight patients with a mixed somatotroph-lactotroph tumor, and in 11 of 26 patients with the pure somatotroph tumor. The meta-analysis showed that the response to cabergoline is dependent on the IGF-1 baseline levels, with greater chances to achieve IGF-1 levels normalization with lower basal IGF-1 levels [53], and not on the presence or absence of hyperprolactinemia.

### 2.6. Acidophil Stem Cell Tumor

These rare tumors consist of a single immature cell type, which is thought to be a common precursor of GH and prolactin cells [13,54]. On histology, these tumors are chromophobic or slightly acidophilic, with abundant granular cytoplasm that is characteristic of oncocytes (Figure 5). They express mainly prolactin but also GH, the latter often focally and/or weakly. Staining for keratins identifies occasional fibrous bodies. The ultrastructural appearance is predominated by the numerous and giant mitochondria [54].

Clinically these tumors usually present with hyperprolactinemia. Hence, in contrast to cases of mammosomatotroph or mixed somatotroph & lactotroph tumors, in the cases of acidophil stem cell tumor, symptoms of hyperprolactinemia are very common, while acromegaly is less frequent [54]. These tumors are characterized by mildly elevated GH levels, and as symptoms of hyperprolactinemia dominate the clinical presentation, the diagnosis of GH excess and acromegaly can be missed if not evaluated properly; this phenomenon has been described as “fugitive acromegaly” [54,55].

Acidophil stem cell tumors generally have a more aggressive behavior than common prolactinomas; these are frequently invasive, fast-growing macrotumors [13,33]. Unlike typical lactotroph tumors, where the degree of hyperprolactinemia is proportional to tumor size, in patients with acidophil stem cell tumors the blood prolactin levels are disproportionally low for the size of the tumor on imaging [14].

From a clinical perspective, these tumors are frequently resistant to dopamine agonists, both in terms of the reduction in prolactin levels and in tumor size. Similarly, in vitro studies have reported resistance to bromocriptine by acidophil stem cell tumors [48,55].

### 2.7. Poorly Differentiated Pit1-Lineage Tumor

The tumor now known as the poorly-differentiated Pit1-lineage tumor, previously known as “silent subtype 3 adenoma”, is a neoplasm composed of poorly-differentiated, polygonal to spindle-shaped chromophobic cells that express Pit1 as well as focally ER and GATA3, and can produce different combinations of GH, prolactin, α-subunit and/or TSH [49,56] (Figure 6). While initially it was believed that these are silent tumors that do not cause hormone hypersecretion, it was later found that these patients may manifest acromegaly, hyperprolactinemia and/or hyperthyroidism [56].

Mete et al. reviewed the epidemiology of 1055 adenohypophysial neuroendocrine tumors from the pathology files of the University Health Network, Toronto. In that series, all Pit1-lineage tumors numbered 316 and represented 29.9% of all surgically resected adenohypophysial tumors; in this group, 44 were poorly-differentiated Pit1-lineage tumors [30].

These tumors are almost always macrotumors, and are more aggressive and invasive, with increased risk for recurrent disease following surgery and lower rates of disease-free survival [10,12]. Radiologic features frequently include cavernous sinus invasion, involvement of the clivus, and frequently both suprasellar and downward growth [56].

While the pathogenesis of these tumors is not well understood, studies have reported that some patients with poorly-differentiated Pit1-lineage tumors were members of MEN1 kindreds [56].

In a detailed series of 25 patients with poorly-differentiated Pit1-lineage tumors and complete radiological and clinical data, there was a residual tumor following pituitary surgery in most cases (65%), with further progression in half of these patients. Complete tumor resection was successful after one pituitary surgery in less than one third of the patients. One patient was treated with dopamine agonist, resulting in a normalization of prolactin levels, but without change in tumor size [56]. One of these tumors ultimately metastasized, representing a pituitary carcinoma [57], but this patient had no clinical features of acromegaly, and the tumor did not express GH.

### 2.8. Pituitary Carcinoma

Pituitary carcinoma is defined as a pituitary tumor with metastatic spread, either craniospinal dissemination or systemic metastases [58,59]. The occurrence of such metastasis is very rare, constituting less than 1% of all pituitary neoplasms [57]. The diagnosis cannot be made on the basis of the primary tumor, as they generally have no distinctive features, resembling other aggressive pituitary tumors until they metastasize. These tumors frequently have high Ki-67 labeling indices, and several reviews have recommended that a Ki-67 level exceeding 10% should raise the suspicion of pituitary carcinoma [57,60,61].

There are several reports of GH-secreting pituitary carcinomas in the literature [61,62,63,64]. A review published in 2011 identified 132 cases of pituitary carcinoma, most of which were functional, most frequently prolactin or ACTH, but only seven tumors (5%) were producing GH [61].

Transcranial or transsphenoidal surgery has an important role in the treatment of pituitary carcinomas, which in many cases requires more than one surgical intervention or surgical treatment of metastases [61]. Other treatment options include medical treatment, which can include one medication or combination treatment, such as dopamine agonists, somatostatin analogs, chemotherapy, temozolomide and others. Radiation is also frequently used in these cases. The different treatment modalities can possibly slow the disease progression [61].

### 2.9. Primary Pituitary Hyperplasia

The diagnosis of pituitary hyperplasia is often made on pathology, based on the presence of intact but expanded pituitary acini that contain all of the adenohypophysial cell types, but with increased numbers of one cell type (Asa 2011); in the case of acromegaly or gigantism, the increase is in somatotrophs and/or mammosomatotrophs. In these cases, as there is diffuse involvement of the pituitary gland, and imaging identifies an enlarged sella with no specific region of gadolinium enhancement that is usually found in the nontumorous gland around a tumor that generally lacks gadolinium enhancement. In some patients with primary pituitary hyperplasia, the disease may progress to multifocal neoplasia; even in the presence of coexisting pituitary tumor(s), these are mostly small tumors, and usually there is no clear evidence of a pituitary tumor on imaging [55].

The diagnosis of pituitary hyperplasia should normally prompt the investigation of a GHRH-secreting tumor (see below), but if none is identified, the diagnosis of primary pituitary hyperplasia should lead the treating physician to consider the possibility of an underlying germline genetic predisposition syndrome, such as MEN1/MEN4, Carney Complex, McCune Albright and X-LAG syndrome [55,65,66,67,68,69].

In patients with MEN1, Carney Complex, and McCune Albright Syndrome there may be evidence of somatotroph/mammosomatotroph hyperplasia with associated pituitary tumor, thus the presence of coexisting hyperplasia and a distinct tumor in a young patient with acromegaly or gigantism should raise the suspicion of a possible familial disease [55].

McCune Albright Syndrome, a genetic but not hereditary disease caused by mosaic somatic mutation in *GNAS*, is associated with acromegaly in 20–30% of cases. Examination of pituitary specimens of these patients revealed that the pituitary disease was diffuse, with hyperplastic and neoplastic changes, suggesting that the primary pathology in cases of acromegaly in these patients is somatotroph hyperplasia, that involves the entire pituitary gland, sometimes with progression to neoplasia [67,69]. In light of the diffuse involvement of the pituitary gland, pituitary surgery in these cases may not be the best treatment option, as it may require total hypophysectomy. For that reason, in patients with acromegaly secondary to McCune Albright Syndrome, medical treatment is preferred, and somatostatin analogs improve GH and IGF-1 levels in most cases, though IGF-1 normalization is not frequent and additional treatment with pegvisomant is usually needed and effective. Of note, when MRI is not available or not feasible, and a CT scan is used, fibrous dysplasia of the sphenoid bone may impair the visualization of a pituitary tumor. Importantly, fibrous dysplasia of the skull base, especially with involvement of the sphenoid bone, may make surgery technically difficult, with an increased risk of hemorrhage due to the high vascularity of fibrous dysplasia [69].

There is a high risk of malignant transformation of fibrous dysplasia in these patients, therefore pituitary irradiation is not recommended [69].

Pituitary hyperplasia is more common in patients with MEN1, due to germline mutation of the *MEN1* gene, or MEN4, secondary to mutation in *CDKN1B*, compared to non-MEN pituitary lesions [66]. Of note, Trouillas et al. reported that multiple tumors are more common in patients with MEN, and may be difficult to distinguish from pituitary hyperplasia on imaging [66].

The Carney complex, resulting from inactivating a mutation in the type 1α regulatory subunit of protein kinase A (the *PRKAR1A* gene), may be associated with acromegaly attributed to pituitary somatotroph/mammosomatotroph hyperplasia that may progress to multifocal neoplasia [67].

X-LAG, resulting from mutation in the *GPR101* gene that encodes an orphan G protein-coupled receptor, usually presents in the first year of life and certainly before four years of age. In these cases the patients frequently develop pituitary hyperplasia (25%) or a bihormonal pituitary tumor, producing both GH and prolactin. A possible cause for early-onset pituitary hyperplasia in these cases is prenatal exposure to increased GHRH levels. In these patients, acromegaly/gigantism presents very early with excessive GH, frequently associated with high prolactin levels. In many cases, multimodal treatment is required, including surgery and radiotherapy. Somatostatin analogs are usually ineffective, and while dopamine agonists can control prolactin levels, no significant effect was evident on GH and IGF-1 levels. Pegvisomant can be used with good results and is of particular importance in patients with pituitary hyperplasia, where extensive surgery is not feasible due to the risk of hypopituitarism [68].

## 3. Extra-Pituitary Tumors

### 3.1. Ectopic Growth Hormone Hypersecretion

Ectopic secretion of GH from extra-pituitary tumors, causing less than 1% of acromegaly cases, was documented in case reports describing patients with bronchial or pancreatic neuroendocrine tumors and lymphomas [70,71,72,73]. There are also reports of ectopic acromegaly due to a GH-secreting pituitary tumor in the sphenoid sinus [74,75].

Clinically, patients present with the typical signs and symptoms of acromegaly and no biochemical features can distinguish ectopic from eutopic GH-secreting tumor [75]. In addition, these patients may manifest additional symptoms from the primary tumor.

In these rare cases, pituitary MRI may be entirely normal, or may show empty sella; there is no evidence of pituitary tumor or pituitary enlargement on imaging, and yet the patient has all the clinical and biochemical features of acromegaly [71,75].

The treatment options in these cases include surgical intervention to remove the primary tumor, and dopamine agonists and somatostatin analogs can also be useful. Ramirez et al. reported a case of a GH-secreting tumor in the sphenoid sinus that was treated surgically, but required adjuvant treatment and had good response to the combination of a somatostatin analog and a dopamine agonist [75].

### 3.2. Excess Production of Growth Hormone-Releasing Hormone

Hypothalamic tumors known as gangliocytomas are very rare tumors that may produce excess GHRH; these benign, slow-growing, neuronal neoplasms are usually diagnosed in children and young adults. Puchner et al. reported that acromegaly was found frequently in the cases of sellar gangliocytoma; in this setting, the neuronal tumor was associated with somatotroph proliferations, either hyperplasia or, most often, sparsely-granulated somatotroph tumors [4,5].Ectopic GHRH secretion, causing pituitary somatotroph hyperplasia, is a rare cause of acromegaly, responsible for less than 1% of cases [3,65,76]. This phenomenon has been described in neuroendocrine tumors of pancreas or lung and pheochromocytomas; some of the patients had had familial syndromes, such as MEN1 [77,78]. Even with prolonged stimulation, there is usually hyperplasia that is thought to be reversible [79,80], however transition to the pituitary tumor has been reported in a case where there was metastasis of the lesion to the pituitary [81].

The diagnosis should be considered based on imaging, as in these cases, the pituitary gland may appear hyperplasic and enlarged with no distinct pituitary tumor on MRI [75,82]. The symmetrical enlargement of the gland has no localized gadolinium enhancement that usually highlights the nontumorous gland and distinguishes it from the less vascular tumor [55].

The finding of pituitary hyperplasia should precipitate investigations to localize an extra-pituitary neuroendocrine tumor by CT, octreotide scintigraphy or 68Gallium DOTATATE-PET/CT scans.

Measuring plasma GHRH can be used to confirm excess GHRH levels [76]. Of note, GHRH is not available as a routine lab test, but is advisable when available. Garby et al. reported that of 20 tumors responsible for the ectopic secretion of GHRH, 12 cases (60%) were pancreatic neuroendocrine tumors, and most of them had distant metastases, seven patients (35%) had bronchial neuroendocrine tumors, and one patient (5%) had an appendiceal well-differentiated neuroendocrine tumor [65]. GHRH can be identified by immunohistochemistry in these tumors. The overall prognosis of these patients with GHRH-secreting tumors is excellent, with 85% survival at five years, as these tend to be well-differentiated neuroendocrine tumors [65].

The main treatment options include surgery, which can induce prolonged remission when a complete tumor resection is possible, and somatostatin analogs. Treatment with somatostatin analogs can directly inhibit GHRH secretion from tumor cells and lead to a reduction in GHRH, GH and IGF-1 levels. Dopamine agonists can also suppress GH levels in these patients [76].

## 4. Conclusions

This review highlights the fact that acromegaly is not a single disorder. There are many types of acromegaly, and some patients with growth hormone excess develop gigantism. The clinical spectrum of this disease varies from florid large stature and disfigurement to subtle features that may not initially appear to be within the framework of this disease; indeed, the diagnosis may be missed in some patients who are thought to have clinically non-functioning pituitary tumors, or may not even be suspected to have a pituitary disorder at all. Patients with chronic dental problems, osteoarthritis, sleep apnea and other seemingly unrelated problems may be undiagnosed for many years. It behooves the clinician to be more aware of these subtle features of GH excess in order to consider the diagnosis.

The various histopathological tumors that give rise to acromegaly provide an explanation for the different clinical, biochemical and radiological characteristics of patients with acromegaly. The different tumor types have distinct pathogenetic mechanisms that can shed light on the variable response to different treatment modalities in this population. This information should be considered in every patient, as it will prove to be useful in the era of precision medicine, in which physicians tailor the correct treatment modality to the right patient.

## Figures and Tables

**Figure 1 jcm-08-01962-f001:**
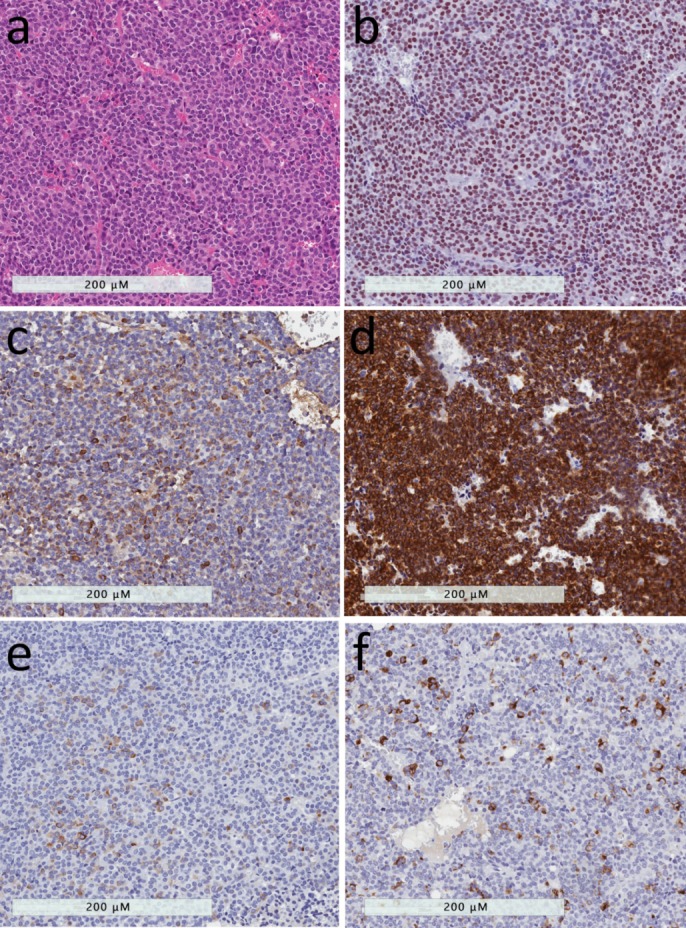
Densely-granulated somatotroph tumor and mammosomatotroph tumor. (**a**) The presence of numerous secretory granules in these tumors results in an eosinophilic appearance that resembles that of the acidophilic cells of the adenohypophysis. (**b**) The tumor cells have intense nuclear reactivity for Pit1. (**c**) They contain abundant growth hormone reactivity. (**d**) The tumor cells have strong and diffuse positivity for low molecular weight cytokeratins using the Cam 5.2 antibody. (**e**) These tumors usually have cytoplasmic positivity for alpha-subunit of glycoprotein hormones. (**f**) The only difference between densely-granulated somatotroph tumors and mammosomatotroph tumors is the expression of prolactin in the latter.

**Figure 2 jcm-08-01962-f002:**
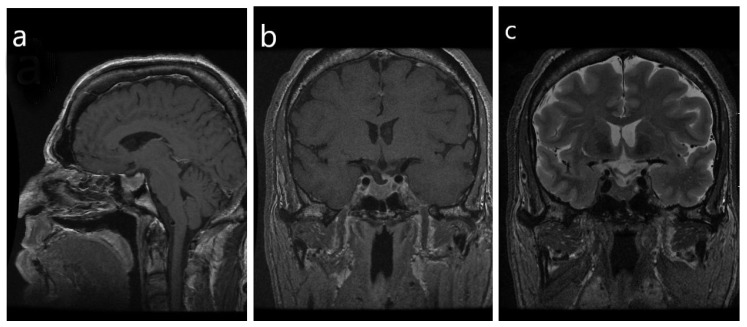
Radiologic imaging of a densely-granulated somatotroph tumor. (**a**) Note the frontal bossing, prognathism and thickening of the scalp with a “rug sign” that are evident on the sagittal view; (**b**) in the coronal view the sella is enlarged, with evidence of a macrotumor. The infundibulum is deviated to the left, with an associated focal defect versus a depression in the sellar floor and an inferior herniation of pituitary glandular tissue. There is ptosis of the optic chiasm, with no evidence of mass effect on the chiasm, (**c**) The tumor does not have a high T2 signal, however, a hypointensity signal is noted within the optic chiasm and proximal cisternal segments of the bilateral optic nerves.

**Figure 3 jcm-08-01962-f003:**
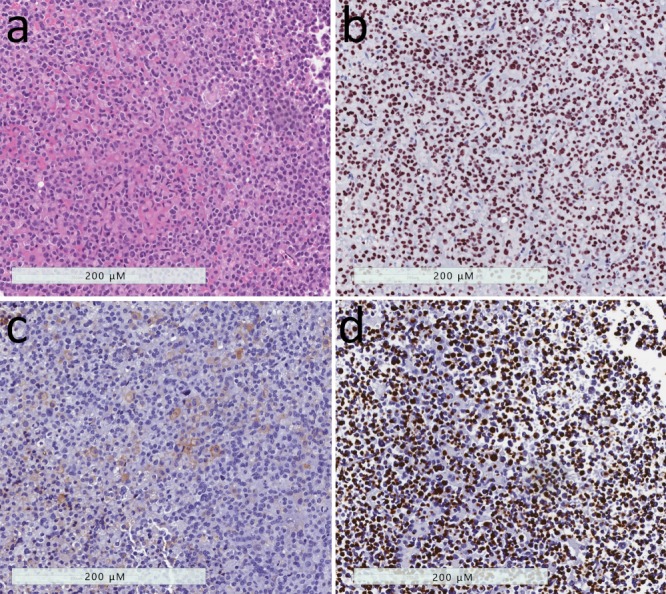
Sparsely-granulated somatotroph tumor. (**a**) These tumors are composed of chromophobic cells with marked nuclear pleomorphism that is attributed to the nuclear distortion induced by large pale globular structures in the tumor cell cytoplasm. (**b**) The nuclei are highlighted by staining for Pit1 that demonstrates the irregular nuclear contours. (**c**) Growth hormone reactivity is scant and faint. (**d**) The tumor cells exhibit a peculiar reactivity for keratins using the Cam 5.2 antibody; the keratins aggregate into round dense fibrous bodies that are the hallmark of this tumor type.

**Figure 4 jcm-08-01962-f004:**
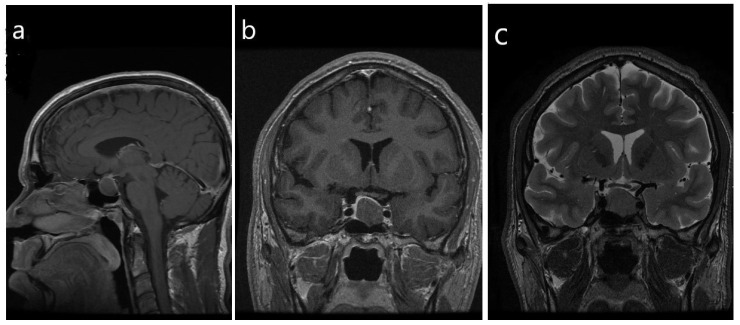
Radiologic imaging of a sparsely-granulated somatotroph tumor. (**a**,**b**) There is a homogeneous hypoenhancing macrotumor that measures 1.5 × 1.7 × 1.9 cm, with mild suprasellar extension and mild mass effect on the optic chiasm. There is mild abutment of the left cavernous ICA, consistent with Knosp grade 1; Note the lack of features of florid acromegaly compared with Figure 1. (**c**) T2-weighted imaging, coronal view reveals that the tumor is homogeneous and isointense to the gray matter.

**Figure 5 jcm-08-01962-f005:**
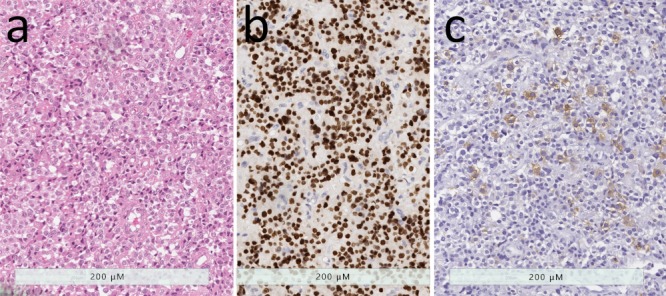
Acidophil stem cell tumor. (**a**) This oncocytic tumor is characterized by an abundant granular cytoplasm that is punctuated by large clear globules, representing dilated swollen mitochondria. (**b**) These tumor have strong nuclear Pit1 reactivity. (**c**) While prolactin is usually diffusely positive (not shown), staining for growth hormone is usually only focal and weak.

**Figure 6 jcm-08-01962-f006:**
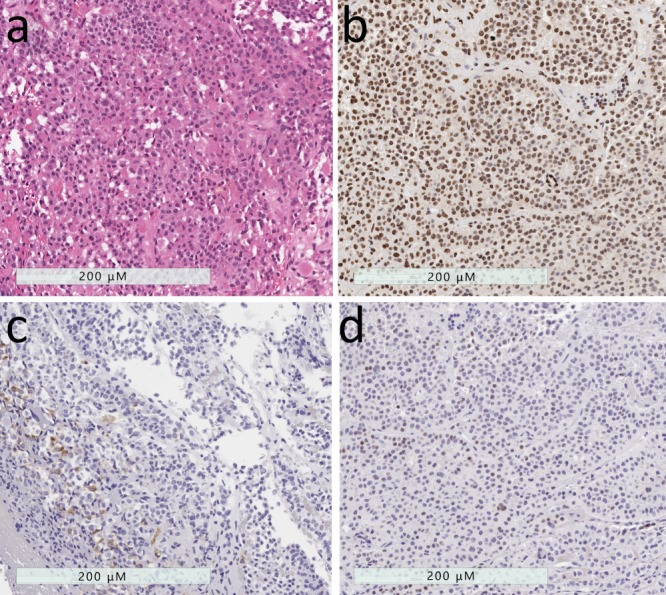
Poorly-differentiated Pit1-lineage tumor. (**a**) These tumors have variable histology, usually formed of polygonal to spindle-shaped cells. (**b**) They have strong nuclear Pit1 reactivity. (**c**) Immunohistochemistry usually reveals variable staining patterns with scattered positivity for GH, PRL, TSH and alpha-subunit of glycoprotein hormones (shown). (**d**) These tumors usually also have variable expression of estrogen receptor and GATA-3.

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
