# Peer review of "The Clinicopathological Spectrum of Acromegaly"

_jcm, 2019, doi:10.3390/jcm8111962_

Round 1

Reviewer 1 Report

The narrative review “The Clinicopathological Spectrum of Acromegaly” from Akirov A. and coworkers is well done, precise and exhaustive. and of scientific interest, since it summarizes pathological profiling, radiological characteristics and clinical behaviour of acromegaly.  Is useful for clinicians to promote personalized approach of this disease.

Despite, similar papers and books have address this issue, in my opinion this paper summarize all the information more focused in a clinician approach and in and easily and helpful manner.

Minor comments:

Line 19:  please, put all in the same font

Line 436: Please,  add  GHRH it is not available as routine lab test but is advisable or strong recommended .

Author Response

Response to Reviewer #1

1) " Line 19: please, put all in the same font "

 Response: We have corrected this now

2) " Please, add GHRH it is not available as routine lab test but is advisable or strong recommended"

Response: we have added this note (lines 430-431)

Sincerely,

Amit Akirov, MD.

Reviewer 2 Report

The authors present the clinicopathological spectrum of acromegaly following an in-depth review of the literature. They show that acromegaly can be caused by a broad spectrum of different diseases. It becomes clear that the various subtypes of GH-secreting pituitary adenomas  are different in terms of patients' clinical presentation, response to therapy and outcome. The content of the article is helpful for all specialities that are involved in the treatment of patients with acromegaly. It is immediately obvious that the manuscript is written by experts in the field and it is a great pleasure to read the manuscript.

Author Response

Thank you for the kind review.